# Complex Autism Spectrum Disorder with Epilepsy, Strabismus and Self-Injurious Behaviors in a Patient with a De Novo Heterozygous *POLR2A* Variant

**DOI:** 10.3390/genes13030470

**Published:** 2022-03-07

**Authors:** Daniel R. Evans, Ying Qiao, Brett Trost, Kristina Calli, Sally Martell, Steven J. M. Jones, Stephen W. Scherer, M. E. Suzanne Lewis

**Affiliations:** 1Department of Family Practice, University of British Columbia (UBC), Victoria, BC V8R 1J8, Canada; dre542@mun.ca; 2Medical Genetics, University of British Columbia (UBC), Vancouver, BC V6H 3N1, Canada; yqiao@mail.ubc.ca (Y.Q.); kcalli@mail.ubc.ca (K.C.); sally.martell@ubc.ca (S.M.); sjones@bcgsc.ca (S.J.M.J.); 3BC Children’s Hospital Research Institute, Vancouver, BC V6H 3N1, Canada; 4iTARGET Autism, Vancouver, BC V6H 3N1, Canada; 5The Centre for Applied Genomics and McLaughlin Centre, Hospital for Sick Children, University of Toronto, Toronto, ON M5G 0A4, Canada; brett.trost@sickkids.ca (B.T.); stephen.scherer@sickkids.ca (S.W.S.); 6Michael Smith Genome Sciences Centre, Vancouver, BC V6H 3N1, Canada

**Keywords:** *POLR2A*, autism spectrum disorder, neurodevelopmental disorder with hypotonia and variable intellectual and behavioral abnormalities (NEDHIB), variant

## Abstract

Autism spectrum disorder (ASD) describes a complex and heterogenous group of neurodevelopmental disorders. Whole genome sequencing continues to shed light on the multifactorial etiology of ASD. Dysregulated transcriptional pathways have been implicated in neurodevelopmental disorders. Emerging evidence suggests that de novo *POLR2A* variants cause a newly described phenotype called ‘Neurodevelopmental Disorder with Hypotonia and Variable Intellectual and Behavioral Abnormalities’ (NEDHIB). The variable phenotype manifests with a spectrum of features; primarily early onset hypotonia and delay in developmental milestones. In this study, we investigate a patient with complex ASD involving epilepsy and strabismus. Whole genome sequencing of the proband–parent trio uncovered a novel de novo *POLR2A* variant (c.1367T>C, p. Val456Ala) in the proband. The variant appears deleterious according to in silico tools. We describe the phenotype in our patient, who is now 31 years old, draw connections between the previously reported phenotypes and further delineate this emerging neurodevelopmental phenotype. This study sheds new insights into this neurodevelopmental disorder, and more broadly, the genetic etiology of ASD.

## 1. Introduction

ASD describes a group of complex neurodevelopmental disorders. ASD is a clinical diagnosis established by the DSM-5 criteria, and characterized by deficits in social communication and interaction, as well as restrictive, repetitive behavior patterns [1]. The Canadian prevalence of ASD is 1 in 66 children [2]. ASD is a heterogenous disorder, with a multifactorial etiology involving both environmental and genetic factors. The genetic contribution to ASD is substantial, with heritability estimated to be between 64% to 91% [3]. Both monogenic and complex genetic traits have been implicated in ASD.

Genes implicated in ASD are involved in a variety of biological pathways, particularly in brain function and development, synaptogenesis, transcription regulation and chromatin remodeling [4,5,6,7]. There is genetic overlap between ASD and other neurodevelopmental disorders such as schizophrenia, epilepsy and intellectual disability [8]. The importance of chromatin remodeling, transcriptional regulation and alternative splicing in ASD has been proposed by prior studies [9,10].

The RNA polymerase II (pol II) complex is a well-studied and essential enzyme responsible for transcribing all protein-coding and some non-coding genes [11]. The largest subunit of pol II is RPB1 which is encoded by *POLR2A*. RPB1 and other subunits form the DNA-binding domain of pol II [12]. The RPB1 trigger loop region facilitates incorporation of incoming nucleotides during the elongation cycle, and residues within the trigger loop have been shown to control the rate of RNA synthesis [13,14].

Recently, *POLR2A* was implicated in an emerging neurodevelopmental disorder abbreviated NEDHIB (Neurodevelopmental Disorder with Hypotonia and Variable Intellectual and Behavioral Abnormalities) (OMIM # 618603). NEDHIB was described by Haijes and colleagues who identified a cohort of 16 patients with ultra-rare de novo *POLR2A* variants ascertained through GeneMatcher [15]. Their analysis yielded 11 probably disease-causing *POLR2A* variants, 4 possibly disease-causing variants and 1 variant was unresolved. Individuals with these variants demonstrated a suite of phenotypic features; the most striking of which was profound early onset general hypotonia and delayed developmental milestones. Additional features included poor speech, intellectual impairment, seizures, strabismus and others. In vitro variant modeling showed that RPB1 mutants introduced into *Saccharomyces cerevisiae* (*S. cerevisiae*) with genetic backgrounds lacking transcription factors *dst1* and *sub1* resulted in aberrant growth; suggesting reduced transcriptional fidelity [15].

Subsequently, Hansen and colleagues identified deleterious *POLR2A* variants in a cohort of 12 individuals ascertained through an online data lake, a pediatrics clinic and an online community for affected individuals [16]. Their analysis included one individual who was previously reported by Haijes and colleagues [15]. Hansen and colleagues identified a higher proportion of epilepsy and a lower proportion of hypotonia in their cohort. Moreover, they described other features including ataxia, joint hypermobility, short stature, skin abnormalities and cardiac congenital anomalies; further expanding the potential phenotypic spectrum.

Here, we describe the phenotype of a patient with confirmed ASD who was further assessed genetically through the Autism Spectrum Interdisciplinary Research (ASPIRE) Program based at BC Children’s Hospital Research Institute in British Columbia (BC), Canada. Genetic investigation using whole genome sequencing of parent–offspring trios revealed a novel de novo heterozygous variant in *POLR2A* (NM_000937.5: c.1367T>C; NP_000928.1: p. Val456Ala). This variant is deleterious according to multiple in silico tools and is not reported in population databases. We describe the phenotype in our patient, draw connections between previously reported phenotypes and further delineate this emerging neurodevelopmental disorder. Our study further emphasizes the phenotypic spectrum of disorders characterized by deleterious *POLR2A* variants and emphasizes the role of this gene in ASD.

## 2. Materials and Methods

### 2.1. Patient Recruitment

The patient presented to the Provincial Medical Genetics Programme (PMGP) Clinic in British Columbia, Canada for assessment of ASD and was invited to participate for research-based genomic testing through the ASPIRE Program [17]. The patient and her parents provided their respective informed consent to participate. The patient was assessed by M.E.S.L. and detailed clinical genetic phenotyping beyond the pre-existing ASD psychometric indices was also performed (see Results).

### 2.2. Psychometric Analysis of ASD

The patient was formally diagnosed with ASD prior to the research study at age 5. Psychometric analyses for ASD were performed using Childhood Autism Rating Scale (CARS). Cognitive and learning assessments were performed using Wechsler Preschool and Primary Scale of Intelligence (WPPSI) and Leiter. Language was assessed using Peabody picture vocabulary test. Adaptive skills and behavior were assessed using the Vineland and Beery test of visual motor integration, according to the Diagnostic and Statistical Manual of Mental Disorders, fourth edition (DSM IV).

### 2.3. Whole Genome Sequencing Pipeline and Variant Analysis

DNA was extracted from whole blood using standard ethanol-based protocols. Parent–offspring trio analysis using whole genome sequencing was performed through ASPIRE’s iTARGET Autism Initiative (www.itargetautism.ca/, accessed on 1 March 2019) in collaboration with the Center for Applied Genomics (TCAG; Sick Kids Hospital, Toronto, Ontario, Canada) and the MSSNG project (Autism Speaks; www.mss.ng/, accessed on 1 April 2018). The pipeline for whole genome sequencing (WGS) has been described elsewhere [18] and is briefly summarized. Whole genome sequencing was performed using Illumina HiSeq X WGS platform by the TCAG at the Hospital for Sick Children. Raw data were aligned to the human reference genome (GRCh38). The depth of WGS was 30x. Variants were detected by importing vcf and bam files into VarSeq (GoldenHelix, Inc., Bozeman, MT, USA, https://www.goldenhelix.com, accessed on 7 July 2020) for (single nucleotide variants (SNVs) and insertion-deletions (indels) and copy number variant (CNV) analysis. CNVs were generated with VarSeq using CNValgorithm with a minimum 10 kb binned region coverage. SNVs/Indels were filtered using quality controls prior to variant annotation with VarSeq. Quality control criteria included read depth ≥ 10 and genotype quality ≥ 20. For homozygous recessive and compound heterozygous variants, a minor allele frequency (MAF) threshold of ≤ 0.05 was selected. Alternatively, a MAF ≤ 0.01 was applied for de novo variants, as well as X-linked variants, and variants in candidate ASD genes, imprinted genes, loss of function variants, as well as those in the 59 actionable ACMG genes, and finally, those in custom in-house gene lists of candidate genes involved in ASD, ID, seizure, hearing loss, overgrowth and others. Variant interpretation was performed using a custom internal pipeline using VarSeq which incorporates data from over 20 different databases. The in silico tools used in our analysis included Sorting Intolerant from Tolerant (SIFT) [19], Polyphen-2 [20], MutationTaster [21], Functional Analysis through Hidden Markov Models (FATHMM) [22], Combined Annotation Dependent Depletion (CADD) [23], Genomic Evolutionary Rate Profiling (GERP) [24] and Grantham scores [25]. The WGS data from this study are available online (https://research.mss.ng/, accessed on 7 July 2020).

### 2.4. Dynamic Molecular Simulations

The amino acid sequence of the protein product of wild-type *POLR2A* (NCBI accession number NP_000928.1), as well as the same sequence with the p.Val456Ala or p.Ile457Thr variant, were used as input to AlphaFold v2.0.1 (DeepMind Technologies Limited, London, United Kingdom) [26] in order to predict their three-dimensional structures. The two predicted structures were visualized using PyMol v2.5.2 (Schrödinger, New York, NY, USA) [27] after superposing them using the “align” command. The predicted structures were also used as input to the molecular dynamics web server CABS-flex 2.0, accessed on 27 February 2022 [28] to assess whether any gross changes in amino acid contacts were predicted. We searched gnomAD [29] to identify common (allele frequency > 1%) missense variants within 20 amino acid residues of the mutation of interest (residues 436–476) that could be used as controls in our structural comparisons, but none were found, consistent with the high missense constraint observed in *POLR2A* (gnomAD missense Z-score = 7.13; observed/expected = 0.42).

## 3. Results

The proband is a female who was born at 41 weeks via induction to a healthy 34-year-old mother. The prenatal and perinatal histories were uncomplicated. Her postnatal history was unremarkable until 18 months, when she suffered a febrile seizure. She was diagnosed with a petit mal seizure disorder and was treated with anticonvulsant therapy from age 5 through to age 8, after which medications were discontinued given a seizure-free period. Her developmental milestones were considered delayed relative to her siblings; however, they were generally achieved within the upper limit of normal for age. She walked at 17 months. Her speech was late with first words occurring after two years, with only babbling prior. She began linking words together after 2 and a half years. She developed self-injurious behaviors primarily manifesting as skin picking, without any head banging or hair pulling. At age 3 she developed temper tantrums when people moved into her physical space. She had selective sensitivity to certain sounds and demonstrated stereotyped movements, facial motor tics and marked sensitivity to touch. Given concerns surrounding dependency, communication, adaptive skills, self-injurious behaviors and hyperacusis, she underwent psychometric testing which identified a definitive diagnosis of ASD at age 5.

Psychometric testing using CARS demonstrated ASD in the mild to moderate range. WPSSI1 was limited due to lack of cooperation and apparent inability to comprehend task requirements. She scored in the borderline range for object assembly. Subsequently, WPSSI2 testing revealed uneven performance subtests. Geometric designs and mazes, both of which involve graphomotor components, were not administered due to fine motor delays. Testing on the verbal scale was discontinued due to severe communication difficulties.

The proband was assessed in the PMGP medical genetics clinic at age 13. Her height was 163 cm (75th percentile), weight was 59.5 cm (75–90th percentile) and head circumference was 56.4 cm (97th percentile; +1.91 standard deviations and proportionate). Both her parents were macrocephalic (> 98th percentile). This was felt to represent familial benign macrocephaly in the overall context of the investigations below [30]. She had no obvious dysmorphisms aside from a slight right-sided facial prominence (Figure 1). She had significant far sightedness secondary to strabismus (intermittent left exotropia) which was corrected with glasses by age 13.

Routine genetic investigations including karyotype, fragile X testing, chromosomal microarray and targeted fluorescence in situ hybridization (FISH) for 22q11 and 22q13 were normal. Brain magnetic resonance imaging (MRI) demonstrated grossly normal structures and several foci of increased white matter intensities, consistent with Virchow-Robin spaces and considered a variant of normal.

The patient and her parents subsequently enrolled in the ASPIRE iTARGET Autism Initiative to elucidate the genetic etiology of her complex ASD phenotype. Whole genome sequencing of the parent–offspring trio uncovered a de novo missense variant in *POLR2A* (NM_000937.5: c.1367T>C; NP_000928.1: p.Val456Ala; chr17:7499070). The *POLR2A* p.Val456Ala variant is deleterious according to SIFT, Polyphen-2 and MutationTaster while FATHMM and MutationAssessor are discordant (Table 1). A highly deleterious effect is predicted by the CADD score (29.6). Furthermore, the variant is novel, absent in gnomAD and ClinVar. The impacted amino acid residue is situated within an active site in the RPB1 domain 2 [14]. Our filtering pipeline examined published candidate ASD genes based on high-ranking scores within the SFARI database, and identified three variants with high SFARI scores, but unconvincing evidence for pathogenicity (Appendix A).

We used AlphaFold [26] to predict the structures of both the wild-type and mutant protein and then visualized the superposed structures using PyMol [27]. We observed no gross change in the three-dimensional structure, either overall (Appendix A) or in the vicinity of the mutation (Appendix A). Further support for this interpretation was derived by visualizing amino acid contacts using the CABS-flex 2.0 server [28], with no gross changes in contacts observed between the two structures (Appendix A). A damaging impact of the mutation may instead be due to weakened hydrophobic interactions in the catalytic pocket due to the decreased side chain bulk of alanine relative to valine, such as between residue 456 and a nearby hydrophobic residue, L505 (Appendix A).

The patient was seen in follow up at age 31. She remains healthy with a high level of functioning and has had no recurrence of seizures. Her intermittent left exotropia eventually led to a progressive deterioration in her distance vision such that she elected to have surgical recession of the left medial and lateral extraocular rectus muscles at age 24, without any complications. There were no further co-morbidities aside from anxiety and depression. Today she is living with family, enjoying a good quality of life and is independent to activities of daily living. She recently obtained her driver’s license. Through assistance with an individualized education program, she completed grade 12 equivalency. She briefly enrolled in college before opting to work a part-time occupation instead. Targeted review of systems, focusing on key clinical features identified by Hansen and colleagues [16] were negative for ataxia, joint hypermobility, skin abnormalities, recurrent fever of unknown etiology, congenital heart disease, immune dysfunction or developmental dysplasia of the hip.

## 4. Discussion

This study explores the genetic etiology of a patient who presented with complex ASD involving epilepsy, delayed communication, self-injurious behaviors and strabismus. Genomic investigation identified a novel de novo *POLR2A* variant (p.Val456Ala), which resides in the catalytic domain of RPB1 and is predicted to be deleterious by multiple in silico tools. Prior functional studies modelling de novo *POLR2A* variants in *S. cerevisiae* demonstrate a reduced fitness thought to be caused by malfunctioning pol II enzyme [15]. Our report provides unique insights into the emerging phenotypic spectrum caused by de novo deleterious *POLR2A* variants. This refines our understanding of the underlying structure and function of pol II and expands our understanding of the variable presentation of this neurodevelopmental phenotype. Our case report demonstrates a patient with milder features compared to existing literature and provides helpful health guidance, phenotype and management insights over a 31-year natural history thus far.

Table 1 shows that the novel p.Val456Ala *POLR2A* variant is deleterious according to SIFT, PolyPhen-2 and MutationTaster. Although FATHMM and MutationAssessor were discordant with these predictions, the high CADD score of 29.6 argues for the deleterious nature of this variant. CADD scores above 20 indicate a variant is among the top 1% of the most deleterious substitutions that can occur in the genome; whereas a score of 30 indicates a variant is within the top 0.1% of the most deleterious substitutions that are predicted to occur [23]. This variant at position 456 resides in the RPB1 domain 2, which is a functionally important domain containing an active site [14]. The valine at this position shows a high degree of evolutionarily conservation as indicated by the GERP score of 5.93. Our dynamic molecular simulations did not show any gross change in the three-dimensional structure resultant from this variant (Appendix A). Valine and alanine are both non-polar amino acids of similar structure as demonstrated by Grantham score. We propose that a damaging impact could arise due to weakened hydrophobic interactions in the catalytic pocket resulting from the decreased side chain bulk of alanine relative to valine. Support for this is demonstrated by molecular dynamic modeling of the adjacent residue (p.Ile457Thr) (Appendix A) which has previously been reported as deleterious by Haijes and colleagues, which also does not grossly alter the protein structure, but results in aberrant growth in *S. cerevisiae* with background *dst1* and *sub1* knockout [15]. We therefore propose future in vitro studies could explore whether this p.Val456Ala alters transcription efficiency. Finally, p.Val456Ala is novel and not previously reported in GnomAD or ClinVar.

Interestingly, there is a patient reported by Haijes and colleagues with a *POLR2A* variant in the active site at the adjacent amino acid residue (p. Ile457Thr) (individual 2) [15]. A notable contrast can be drawn between this individual and the one described in our study, as they were more severely affected compared to our patient. For example, individual 2 was hospitalized on day 3 of life for feeding difficulties and cyanotic spells with oxygen desaturations. He was followed by physiotherapy at 7 weeks of age for general hypotonia and he developed three respiratory tract infections requiring antibiotics by 8 months. He was managed in a tertiary care center for developmental delays. He had slow development in gross and fine motor skills, dystrophic muscle mass and significant muscle weakness. He sat without support at 23 months, walked by > 55 months and had MRI changes showing delayed myelination and wide ventricles. There were no concerns for autistic behaviors reported by 4 years of age. This contrasts with the phenotype presented in our patient, which was predominantly characterized by complex ASD with epilepsy and strabismus, and behavioral abnormalities. Her milestones were within an upper limit of normal (walking at 17 months), and she primarily experienced speech delays, which gradually improved. At age 31, she is independent to activities of daily living, working part-time and maintains a good quality of life and functional status. To our knowledge, she is the oldest patient reported in the literature with this phenotype, and our detailed clinical information spans a longer interval.

More broadly, our patient presents with a milder phenotype compared to most previously reported individuals with de novo *POLR2A* variants. For example, Haijes and colleagues describe the most striking feature in their cohort was infantile onset hypotonia in 14 out of 16 individuals [15]. Moreover, hypotonia was reported as profound in nine of these individuals. Conversely, they only described two individuals in their cohort without hypotonia. Of these two individuals, one reportedly had autistic behaviors (without evidence of psychometric testing), and little information was available for the second individual. Interestingly, Hansen and co-authors observed a lower prevalence of hypotonia in their cohort (8 out of 12 individuals) compared to Haijes and colleagues’ previously reported cohort with *POLR2A* variants (*p* = 0.076309) [16]. In their study, five individuals were without hypotonia, however full phenotypic information was available for only one individual (individual eight).

This person (individual eight) was a 14-year-old girl who was striking for her mild phenotype compared to other individuals in their cohort. She had seizures and speech delay like our patient, however, her seizure onset was from 13 years old, and she had an abnormal MRI showing polymicrogyria involving large portions of the left cerebral hemisphere among other findings [16]. She had hypertonia (age eight) with spasticity in the right extremities, with motor delays, ataxia, intellectual disability, and no reported autistic features. Thus, her phenotype was more severe than our patient. Remarkably, individual eight inherited the distal *POLR2A* variant (p.Gln1814Valfs99ter) from her mother, who was reported to have speech delay and only mild learning difficulties. The paucity of available clinical information regarding milder phenotypes underscores the importance in detailing the complete phenotypic spectrum of patients with de novo *POLR2A* variants. Milder phenotypes and the potential for inherence of *POLR2A* variants raises important questions about other individuals who could be diagnosed through genetic testing.

The different methods of ascertainment stand out as a distinct feature in our case report compared to previous studies. Our patient presented to the BC PMGP clinic for genetic assessment due to her diagnosis of ASD, whereas previously published cohorts were ascertained through GeneMatcher, Hadoop Architecture Lake of Exomes (HARLEE) or a pediatrics clinic. Four individuals were described with autistic behaviors by Haijes and colleagues [15] while Hansen and co-authors reported six individuals with ASD in their cohort [16]. Unfortunately, information surrounding the autistic features in previous studies are limited by a lack of reported information, such as whether cases had confirmed with psychometric testing and what functional impact these diagnoses had on the patients. Our case identifies a patient with psychometrically confirmed high-functioning ASD diagnosed at age five, which was primarily marked by temper tantrums, dependency and skin picking, without any documented intellectual disability. Re-visited at age 31, our patient enjoys an excellent quality of life as described above. Thus, the milder phenotypic presentation identified in this study could have profound implications with respect to identifying other patients around the world who harbor *POLR2A* variants. This study emphasizes the importance of future studies investigating *POLR2A* variants specifically in the context of ASD. Future studies should explore whether patients with mild autistic features +/− epilepsy in the absence of intellectual disability might harbor *POLR2A* variants.

In conclusion, we identified and characterized a novel heterozygous de novo *POLR2A* variant which provides key insights into an emerging neurodevelopmental phenotype and its clinical variability. Our study and previously published cohorts of patients with *POLR2A* variants demonstrate the diagnostic utility of next-generation sequencing as it pertains to ASD and other neurodevelopmental disorders. These findings expand our understanding of the clinical and genetic heterogeneity of phenotypes caused by *POLR2A* variants and highlight the importance of this gene in the context of contributing to ASD.

## Figures and Tables

**Figure 1 genes-13-00470-f001:**
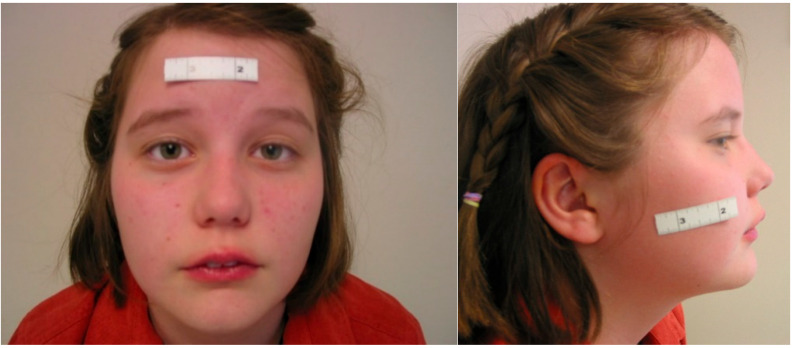
Clinical features of the proband with complex ASD involving epilepsy and strabismus. The photograph depicts the patient aged 13 years old. There are no dysmorphic features aside from subtle right-sided facial prominence. Her strabismus was corrected with glasses (not shown), and eventually was surgically corrected by age 24.

**Table 1 genes-13-00470-t001:** Bioinformatic Tools Predict the *POLR2A* p. V456A Variant is Deleterious.

Tool	Score	Prediction
SIFT	0.01	Deleterious
Polyphen-2 (HDIV)	0.998	Probably damaging
MutationTaster	0.999889	Damaging
FATHMM	−1.09	Tolerated
MutationAssessor	0.94726	Functional
CADD PHRED	29.6	Deleterious
GERP ++ RS	5.93	Conserved
Grantham	64	Similar

## Data Availability

The data presented in this study are available on request from the corresponding author. The data are not publicly available due to confidentiality.

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
