# Peer review of "Complex Autism Spectrum Disorder with Epilepsy, Strabismus and Self-Injurious Behaviors in a Patient with a De Novo Heterozygous POLR2A Variant"

_genes, 2022, doi:10.3390/genes13030470_

Round 1

Reviewer 1 Report

No details are given about the patient’s strabismus such as age of onset and management. The photograph does not show any overt residual strabismus.  The inclusion in the title sets up an anticipation not fulfilled in the results section.

Are there any details about her self-injurious behaviors (hair pulling, head-banging, biting, etc.)? The only clue is in the discussion rather than the results section.

To give a more complete picture of your patient, the details of driving and working part-time (at what level of employment skills) and educational achievement should be in the results.

The patient, referred to in line 219, to whom you compare your case, had her seizure onset at age 13 years (albeit with a mildly abnormal EEG at 8 years of age) and had an abnormal MRI documented at 8 years of age.  Pointing out these differences would strengthen the contrast between your high functioning patient and the majority of cases reported so far.

Hanson et al. indicate that individual 8 inherited the variant from her mother who had mild learning difficulties.  Your use of the term "appears to have inherited" is not supported by the statement in Hansen et al. in which they clearly state that this variant was inherited.  An opportunity was missed to speculate further that the mother of individual 8, in that report, further emphasizes the potential spectrum of the phenotypic variants, heeding the caveat that this variant was the most distal known to date.

This case report also provides an opportunity to establish a position on the role of exome sequencing in the evaluation of patients with autistic spectrum disorders.  There is no overt and unique series of facial characteristics reported by yourselves or the 2 series of POLR2A-associated disease.  The cases of POLR2A-associated neurodevelopmental disabilities have all been ascertained through high-throughput sequencing.  Such sequencing is not considered part of the necessary evaluation for ASD in published guidelines.  While genomic sequencing is still primarily a research tool, exomic sequencing is in widespread clinical use.

Author Response

Thanks

Reviewer 2 Report

The study by Evans et al. describes an individual diagnosed with an autism spectrum disorder in which a de novo point mutation in the POLR2A gene was identified. The variant was predicted to be deleterious by some but not all prediction tools, but no functional validation was provided. This lack of validation may be problematic given reports of macrocephaly (which is commonly related to autistic features) in the proband as well as her both parents (>90 percentile).

Previous studies have characterized a total of 27 patients with ultra-rare de novo POLR2A mutations and validated the deleterious impact of the genetic variations using in vitro model systems. The patient described by Evans et al. presented a relatively mild and somewhat different phenotype, this could potentially be an important observation in the context of POLR2A genetic disorders.

Major concern:

Please indicate how the lack of hypotonia in the proband, the macrocephaly that is also present also in both parents, and the prediction of deleteriousness in some both not all the prediction tools justifies a suggestive causal impact of the observed mutation.

Author Response

Thanks

Reviewer 3 Report

This is an interesting case report; however, it should be improved.

Authors did not clearly elucidate the investigation methods:

- DNA sample collection and processing should be indicated and in silico analysis tools are not described, and these must be added to M&M section.

- Have been molecular dynamic simulations of the mutated protein performed to classify the novel variant as deleterious? Please, explain this in the M&M and results sections.

Results section:

- Image of in silico analysis (protein modelling) representing the wild type and mutate protein, with focus on the amino acid substitution, will helps the reader and should be added.

- Whole genome sequencing of parent-offspring trio offers the possibility to perform the analysis of transmission. Are authors able to perform this analysis? This is of particular interest also to elucidate the genotype.

- The patient can carry also others known ASD causative mutations (i.e. those reported in SFARI database), have authors checked? Can authors provide this analysis?

Furthermore, also discussion can be improved:

- NEDHIB and ASD conditions are not clearly compared in relation to the novel mutation.

- The suggested analysis will offer new data to be discussed.

Author Response

Thanks

Round 2

Reviewer 2 Report

Dear authors, thank you for your replies and the revisions in the manuscript.

I only ask you for minor revisions (concerning point #3):

1. Thank you for clarifying the lack of hypotonia in the manuscript (i.e., some of the patients in previous studies also did not display hypotonia).

2. The mild presentation of macrocephaly in the proband is now sufficiently addressed. The concern (i.e., that possible genetic effects driving both parents into the >98 percentile of head circumference may affect brain development of the proband) remains, but this point is addressed sufficiently.

3. The various in-silico tools are now described in more detail, some indicating a possible deleterious effect of the identified mutation. Related to this, the authors have now also included a 3D model to further strengthen these findings. However, the results indicate no changes in the three-dimensional protein structure. This finding further raises the question of whether the identified mutation is deleterious. As correctly mentioned by the authors, valine and alanine have a very similar structure and polarity, and differ only slightly in hydrophobic interaction based on Grantham scores. Could you please indicate whether other mutations with similar predicted changes are known, that have been validated using functional assays? For example, are there previously validated POLR2A mutations with similar “mild” effects on protein structure?

Author Response

Minor revisions include an analysis of the impact of a functionally validated variant as requested. supp figure added. 

Reviewer 3 Report

Dear colleagues, I appreciate replies and all the revision in the manuscript. I ask you only minor revision.

Methods section:

i) Can you indicate the deep of WGS?

ii) I suggest you deposit the WGS data on data repository, even with controlled access, and to specify in the text the link, if it is possible.

Results section:

In the previous revision: (iii) The patient can carry also others known ASD causative mutations (i.e. those reported in SFARI database), have authors checked? Can authors provide this analysis?

You: There are no known pathogenic variants in ASD associated genes in our analysis. Fragile X testing was negative and there are no pathogenic variants in the ACMG clinically actionable genes (i.e. including PTEN). There are also no de novo variants in the SFARI genes. There are 3 variants in genes with the highest score on SFARI (i.e. RERE, ZMYND11 and HERC2). Each of these are paternally inherited, while both parents in the study family are neurotypical, thus paternal transmission is not in keeping with the family structure. Furthermore, HERC2 is associated with autosomal recessive transmission and thus unlikely to be causative, while RERE and ZMYND11 are OMIM genes with autosomal dominant pattern. Further, the variants are not strong candidates on a variant level. Each of these variants were found in normal populations, albeit in a low frequency (in gnomAD, 16 counts as allele frequency  0.0001057 for the variant in RERE; 9 counts as the allele frequency 0.00005913 for the variant in ZMYND11; and 95 counts as frequency 0.0006241 for the variant in HERC2), compared against the de novo variant in POLR2A, which is absent from gnomAD. Moreover, the POLR2A variant has the highest CADD score and no more than 3 out of 5 bioinformatics tools predicted any other variant as damaging (in fact, only 1 out of 5 bioinformatics tools predicted the variant in ZMYND11 as damaging). While it is possible that variants in these 3 SFARI genes might contribute to the ASD, on balance, the variants identified in this case were poor candidates as compared to the variant in POLR2A, and thus we didn’t consider them for discussion.  

New (iii) I think your answer is interesting, can you add these results and considerations in the manuscript?

Author Response

minor revisions to the methods done and supp. results added as requested. 
